# Integrated Transcriptomics and Metabolomics Analysis of the Fructan Metabolism Response to Low-Temperature Stress in Garlic

**DOI:** 10.3390/genes14061290

**Published:** 2023-06-19

**Authors:** Haiyan Bian, Qianyi Zhou, Zhongping Du, Guangnan Zhang, Rui Han, Laisheng Chen, Jie Tian, Yi Li

**Affiliations:** 1Laboratory for Research and Utilization of Germplasm Resources in Qinghai Tibet Plateau, Academy of Agricultural and Forestry Sciences of Qinghai University, Xining 810016, China; BC20210212@163.com (H.B.); z763335329@163.com (Q.Z.); 1988990022@qhu.edu.cn (Z.D.); 2002990025@qhu.edu.cn (G.Z.); hanrui@qhu.edu.cn (R.H.);; 2Qinghai Key Laboratory of Vegetable Genetics and Physiology, Academy of Agricultural and Forestry Sciences of Qinghai University, Xining 810016, China

**Keywords:** *Allium sativum* L., fructan, transcriptome, metabolome, regulatory network

## Abstract

As the main reserve carbohydrate in garlic, fructan contributes to garlic’s yield and quality formation. Numerous studies have shown that plant fructan metabolism induces a stress response to adverse environments. However, the transcriptional regulation mechanism of garlic fructan in low-temperature environments is still unknown. In this study, the fructan metabolism of garlic seedlings under low-temperature stress was revealed by transcriptome and metabolome approaches. With the extension of stress time, the number of differentially expressed genes and metabolites increased. Using weighted gene co-expression network analysis (WGCNA), three key enzyme genes related to fructan metabolism were screened (a total of 12 transcripts): sucrose: sucrose 1-fructosyltransferase (*1-SST*) gene; fructan: fructan 6G fructosyltransferase (*6G-FFT*) gene; and fructan 1-exohydrolase (*1-FEH*) gene. Finally, two hub genes were obtained, namely Cluster-4573.161559 (*6G-FFT*) and Cluster-4573.153574 (*1-FEH*). The correlation network and metabolic heat map analysis between fructan genes and carbohydrate metabolites indicate that the expression of key enzyme genes in fructan metabolism plays a positive promoting role in the fructan response to low temperatures in garlic. The number of genes associated with the key enzyme of fructan metabolism in trehalose 6-phosphate was the highest, and the accumulation of trehalose 6-phosphate content may mainly depend on the key enzyme genes of fructan metabolism rather than the enzyme genes in its own synthesis pathway. This study not only obtained the key genes of fructan metabolism in garlic seedlings responding to low temperatures but also preliminarily analyzed its regulatory mechanism, providing an important theoretical basis for further elucidating the cold resistance mechanism of garlic fructan metabolism.

## 1. Introduction

*Allium sativum* L., also known as garlic, is a biennial herb of *Allium* in Liliaceae and is the second most important species of *Allium* [1]. Garlic originated in Central Asia [2] and has been widely cultivated worldwide for both medicine and food [3,4]. Garlic seed bulbs can germinate at 3–5 °C, and the seedlings grow rapidly at 12–16 °C. Despite being a cool-season vegetable [5], ‘Ledu Purple Skin Garlic’, an agricultural variety of geographical importance, always suffers frequent cold snaps during the spring in the Qinghai–Tibet plateau. Autumn/winter-sown garlic in the Qinghai province faces low-temperature stress at the seedling stage, which not only leads to the occurrence of physiological abnormalities but also affects the quality and yield of garlic products.

Low-temperature stress is one of the most important abiotic factors limiting plant growth and development. To overcome adversity, plants have evolved several adaptive mechanisms, which will trigger changes in gene expression and subsequently lead to physical and chemical modifications, thereby enhancing their cold tolerance [6,7]. The change in stress-induced gene expression is a key component in the molecular mechanism of plant adaptation to different environments [8]. Many studies have emphasized the importance of transcriptional regulation in plant adaptation to low-temperature stress through transcriptome profiling. For instance, low-temperature stress induced changes in more than 2% of genes in the wheat genome and showed different expression modes [9]. In addition to transcriptomics, metabolomics has been used to analyze defense strategies in various species under low-temperature stress [10]. Moreover, an increasing number of studies have investigated the cold tolerance mechanism of plants using integrated transcriptome and metabolome analysis. Comparing and analyzing the differential expression genes and metabolites of two tobacco varieties under low-temperature stress, Jingjing et al. [11] found that, although most of the cold response genes and metabolites exist in the two varieties, the degree of change was more obvious in the cold-resistant variety, which may be one of the reasons for the difference in cold tolerance between the two varieties.

Fructan is a carbohydrate composed of multiple fructose groups connected by glycosidic bonds [6]. Garlic fructan serves not only as a storage polysaccharide in the tissues [12] but also as a kind of functional food that has good value in promoting digestion and improving immunity [13]. Fructan metabolic regulation is an important protective mechanism for plants to adapt to stress [14]. Many studies have shown that the accumulation of fructan is beneficial for enhancing plant stress resistance. Tognetti et al. [15] showed that under low-temperature induction, the fructan content of winter wheat varieties with strong cold resistance is about three times that of varieties with weak cold resistance. Kawakami et al. [16] found that wheat varieties with strong frost resistance accumulated more fructans than other wheat varieties, indicating that fructan accumulation can effectively improve cold resistance in wheat. Fructans play an important role in plant low-temperature tolerance and the regulation of water uptake [17]. At present, research on garlic fructan mainly focuses on the extraction method [18] and structure [19], and there is a lack of research on its mechanism and regulatory network in a stressed environment. Therefore, the garlic fructan synthase gene needs further exploration. In this study, integrated transcriptomics and metabolomics analysis methods were used to screen the response genes/metabolites of fructan metabolism, construct a regulatory network, and clarify the cold resistance regulatory mechanism of garlic fructan metabolism in garlic under low-temperature stress. This study provides a theoretical basis for garlic resistance breeding in the future.

## 2. Materials and Methods

### 2.1. Plant Materials and Treatment

In this study, ‘Ledu purple skin garlic’ was used as the experimental material. Garlic bulbs of uniform size and released dormancy were selected and seeded in pots. The cultivation medium was German Base Substrate #413 (Klasmann-Deilmann GmbH, Germany Ltd., Niedersachsen, Germany). The pots were placed in a plant light incubator (temperature: 25 °C during the day, 15 °C at night; light: 14 h; relative humidity: 70%) for cultivation until the garlic seedling stage. When the seedling height reached approximately 13 cm, a low-temperature stress treatment was carried out. Half of the seedlings were placed in a low-temperature incubator (4 °C/4 °C, day/night), and the other environmental parameters remained unchanged. At 0 h, 1 h, 3 h, 6 h, 12 h, 1 d, 2 d, 3 d, 4 d, 5 d, 6 d, 9 d, 12 d, and 15 d, the leaves of 5 garlic plants were sampled as biological repeats. The control and low-temperature stress treatments were repeated 3 times. The samples were stored at −80 °C immediately after being frozen in liquid nitrogen.

### 2.2. RNA Extraction and cDNA Synthesis

The total RNA of garlic samples was extracted according to the instructions of the TRNZOL Extraction Reagent (DP 424, Tiangen Biochemical Technology Ltd., Beijing, China). The integrity of the RNA was detected using 1% agarose gel electrophoresis. The quality of the total RNA was detected using a TGem microspectrophotometer (OSE-260, Tiangen Biochemical Technology Ltd., Beijing, China). Part of the extracted RNA was used for real-time fluorescence quantitative PCR, and the other part was used for transcriptome sequencing (completed by Biotechnology Company, Wuhan, China). The synthesis of first-strand cDNA was performed using the above RNA sample as the template, and the FastQuant cDNA First Strand Synthesis Kit (kr106, Tiangen Biochemical Technology Ltd., Beijing, China) was used following the manufacturer’s instructions. The synthesized cDNA was stored in a freezer at −20 °C.

### 2.3. Quantitative Real-Time PCR (qRT–PCR) Analysis

Referring to the research of Minliu et al. [20], a gene suitable for abiotic stress (cyclophilin, *CYP*) was selected as the internal reference gene. The garlic sucrose: sucrose 1-fructosyltransferase gene (*1-SST*) and fructan 1-exohydrolase genes (*1-FEH*) were selected as the target genes. The primers were designed using Primer5.0 software and synthesized by Shenggong Bioengineering Ltd.(Shanghai, China). According to the instructions provided by Superreal fluorescent quantitative premixed reagent (SYBR green, FP205) of Tiangen Biochemical Technology Co., Ltd., the prepared mixture was placed in a lightcycle480 II real-time fluorescence quantitative analyzer for PCR amplification. The reaction system was 20 μL, and the reaction procedure (three steps provided in the manual) was as follows: pre-denaturation at 95 °C for 15 min and 40 cycles of denaturation at 95 °C for 10 s, annealing at 62 °C, and extension at 72 °C for 32 s. Each PCR was repeated three times. The Ct value obtained through qRT-PCR technology was used to calculate the value of 2^−ΔΔCt^, which denotes the relative expression level of the gene.

### 2.4. Construction and Detection of the cDNA Library

RNA extraction and quantification and transcriptome sequencing were conducted using the Illumina HiSeq high-throughput sequencing platform (Illumina, San Diego, CA, USA) by Wuhan MetWare Biotechnology Co., Ltd. (Wuhan, China) following their standard procedures, and the raw data were obtained. Total RNA was isolated from leaves during the seedling stage, and three biological replicates were used. A NanoPhotometer spectrophotometer was used to detect RNA purity (OD260/280 and OD260/230). A Qubit 2.0 fluorometer was used to measure the RNA concentration with high accuracy. An Agilent 2100 biological analyzer was used to determine the RNA integrity value (RIN). The integrity and quality of the RNA were then measured using 1% agarose electrophoresis. Ribosomal RNA was removed from the total RNA to obtain mRNA. Then, a fragment buffer was added to break the RNA into short fragments. First-strand cDNA was synthesized with random hexamers using short-segment RNA as a template, and second-strand cDNA was synthesized by adding buffer, dNTPs, and DNA polymerase I. The double-stranded cDNA was purified using ampurexpbeads. Purified double-stranded cDNA was subjected to terminal repair, and an A-tail was added and sequenced. The fragment size was selected by ampurexpbeads, and the final cDNA library was obtained by PCR enrichment. After construction of the library, qubit2.0 was used for preliminary quantification, and Agilent 2100 was used to detect the insert size of the library. After passing the detection, qRT–PCR was used to accurately quantify the effective concentration of the library (effective concentration of the library > 2 nm) to complete the library inspection. After the library was qualified, different libraries were pooled according to the target offline data volume and sequenced using the Illumina HiSeq platform.

### 2.5. Original Data and Filtering and Transcript Splicing

Before data analysis, the original data were filtered to ensure that these reads were of high quality. These data were used for splicing transcripts to ensure the accuracy of subsequent analysis. When filtering the original data, reads with adapters, N content exceeding 10% of the read alkali base number, and low-quality (Q ≤ 20) alkali base number exceeding 50% of the read alkali base number were removed. After filtering the original data, checking the sequencing error rate, and checking the GC content distribution, Trinity software (v2.6.6) was used to splice the filtered clean reads and corset hierarchical clustering was performed to obtain the longest cluster sequence as a UniGene for subsequent analysis.

### 2.6. Gene Function Annotation and CDS Prediction

The UniGene sequence was compared with the KEGG, NR, Swiss-Prot, GO, COG/KOG, and TrEMBL databases using blast software (v2.7.1). After the amino acid sequence of the UniGene was predicted, Hmmer software (HMMER 3.2 package) was used to compare with the Pfam database to obtain the annotation information of the UniGene. When the UniGene was compared to the protein database (NR, SwissProt, KEGG, KOG), the ORF coding frame information of the transcript was extracted from the comparison results, and the coding region sequence was translated into an amino acid sequence (in the order of 5′→3′), according to the standard codon table. The coding region sequence and corresponding amino acid sequence of the UniGene were predicted by transdecoder software.

### 2.7. Functional Annotation and Enrichment Analysis of Differentially Expressed Genes (DEGs)

The unigenes of different stages were screened by the double difference value and *p*-value (|log2Foldchange| ≥ 1, and FDR < 0.05). Based on the KEGG database (http://www.genome.jp/KEGG/, accessed on 10 April 2019), function and pathway analysis of the genes involved in this study were obtained.

### 2.8. Sample Preparation and Extraction

The freeze-dried leaves were crushed using a mixer mill (MM 400, Retsch) with a zirconia bead for 1.5 min at 30 Hz. One hundred milligrams of powder was weighed and extracted overnight at 4 °C with 0.6 mL 70% aqueous methanol. Following centrifugation at 10,000× *g* for 10 min, the extracts were absorbed (CNWBOND Carbon-GCB SPE Cartridge, 250 mg, 3 mL; ANPEL, Shanghai, China, www.anpel.com.cn/cnw, accessed on 1 May 2019) and filtered (SCAA-104, 0.22 μm pore size; ANPEL, Shanghai, China, http://www.anpel.com.cn/, accessed on 1 May 2019) before UPLC–MS/MS analysis.

### 2.9. UPLC Conditions

The sample extracts were analyzed using a UPLC–ESI–MS/MS system (UPLC, Shim-pack UFLC SHIMADZU CBM30A system, www.shimadzu.com.cn/, accessed on 1 May 2019; MS, Applied Biosystems 4500 Q TRAP, www.appliedbiosystems.com.cn/, accessed on 1 May 2019). The analytical conditions were as follows: UPLC column, Waters ACQUITY UPLC HSS T3 C18 (1.8 µm, 2.1 mm × 100 mm). The mobile phase consisted of solvent A (pure water with 0.04% acetic acid) and solvent B (acetonitrile with 0.04% acetic acid). Sample measurements were performed with a gradient program that employed the starting conditions of 95% A and 5% B. Within 10 min, a linear gradient to 5% A and 95% B was programmed, and a composition of 5% A and 95% B was maintained for 1 min. Subsequently, a composition of 95% A and 5.0% B was adjusted within 0.10 min and maintained for 2.9 min. The column oven was set to 40 °C, and the injection volume was 4 μL. The effluent was alternatively connected to an ESI-triple quadrupole-linear ion trap (QTRAP)-MS.

### 2.10. ESI-Q TRAP-MS/MS

LIT and triple quadrupole (QQQ) scans were acquired on a triple quadrupole-linear ion trap mass spectrometer (Q TRAP), API 4500 Q TRAP UPLC/MS/MS System, equipped with an ESI Turbo Ion-Spray interface, operating in positive and negative ion mode and controlled by Analyst 1.6.3 software (AB Sciex). The ESI source operation parameters were as follows: ion source, turbo spray; source temperature 550 °C; ion spray voltage (IS) 5500 V (positive ion mode)/−4500 V (negative ion mode); ion source gas I (GSI), gas II (GSII), and curtain gas (CUR) were set at 50, 60, and 30.0 psi, respectively; the collision gas (CAD) was high. Instrument tuning and mass calibration were performed with 10 and 100 μmol/L polypropylene glycol solutions in the QQQ and LIT modes, respectively. QQQ scans were acquired as MRM experiments with the collision gas (nitrogen) set to 5 psi. DP and CE for individual MRM transitions were performed with further DP and CE optimization. A specific set of MRM transitions was monitored for each period according to the metabolites eluted within this period.

### 2.11. Screening and Functional Annotation of Differential Expression Metabolites

The differential expression metabolites were screened by combining fold change with variable importance in the project (VIP) of the opls-da model. The screening criteria were fold change ≥ 2 and fold change ≤ 0.5. The VIP value represents the influence intensity of the difference between groups of corresponding metabolites in the classification and discrimination of each group of samples in the model. Based on this, metabolites with VIP ≥ 1 were selected. The KEGG database was used to annotate the obtained differential expression metabolites, and the enrichment degree of the metabolites in each pathway was counted.

### 2.12. Integration of Transcriptome and Metabolome

According to the results of differential expression metabolite analysis and transcriptome differential expression gene analysis, the same group of differential expression genes and differential expression metabolites were mapped to the KEGG pathway map at the same time to better understand the relationship between genes and metabolites. The interaction network of fructan-related differentially expressed genes (DEGs) detected by transcriptome and fructan-related differential expression metabolites measured by metabolomics were analyzed, and an interaction network diagram was constructed with R language to preliminarily reveal the regulatory network of garlic fructan in response to low-temperature stress. The searched DEGs were used to generate co-expression network modules using the weighted gene co-expression network analysis (WGCNA) package, and the obtained co-expression modules were merged on eigengenes [21]. All DEGs were divided into different modules according to their expression patterns. Eigengene values were calculated for each module and used to search for the association with sugar metabolites.

### 2.13. Data Statistics and Analysis

Excel 2010 was used to analyze the experimental data, SPSS 21.0 was used to analyze the significance of the differences, and Origin 2018 was used to generate the figures.

## 3. Results

### 3.1. Selection of Key Periods of Garlic Fructan Gene Expression

qRT–PCR showed the relative expression of *1-SST* and *1-FEH* under normal culture (25 °C) and low-temperature stress (4 °C). As shown in Figure 1, *1-SST* and *1-FEH* were both expressed under normal culture and low-temperature treatment, but before 12 d of treatment, the expression levels of the two genes under low-temperature stress were higher than under the normal expression levels, indicating that low temperatures promoted the response of key fructan enzyme genes at this stage. In addition, under low-temperature stress, *1-SST* gene expression was greater than *1-FEH* expression, which means that fructan synthesis was greater than degradation. Fructan synthesis was the main factor at low temperatures. There was no significant difference in *1-SST* gene expression between the treatment group and the control group from 0 to 12 h, but after 1 d, the expression of the *1-SST* gene in the treatment group was significantly higher than that in the control group. Under low-temperature treatment, the gene expression of *1-SST* showed a trend of first increasing and then decreasing, reaching the highest value at 5 d of treatment. Although the highest value of *1-FEH* gene expression appeared in this period, the expression of the *1-SST* gene was 46.77 times that of *1-FEH*, so it can be regarded as the period with the most fructan synthesis. In summary, 1 and 5 d are the critical periods for key garlic fructan enzyme genes to begin to change and peak under low-temperature stress.

### 3.2. Transcriptome Analysis Results

#### 3.2.1. Sequencing and De Novo Assembly of the Garlic Transcriptome

Total RNA was extracted from garlic leaves treated with three independent biological samples (0 h, 1 d, and 5 d of low-temperature stress). RNA Quality Detection showed that the 5S, 18S, and 28S rRNA bands were intact without obvious tailing, indicating that the extracted RNA had suitable integrity (RIN value). The RNA concentration and quality were determined, and the A260/A280 and A260/A230 of RNA were between 1.9 and 2.1 and between 2.1 and 2.3, respectively. Nine cDNA libraries were constructed and sequenced using Illumina. The total number of clean reads obtained by sequencing was 56.18 Gb, and the clean reads of each sample reached 5.7 Gb. The quality statistics of the clean reads are shown in Appendix A. Q20 (the proportion of base number with a mass value greater than 20 to the total base number) was higher than 98%, and the proportion of Q30 was higher than 95%. The percentage of G and C bases in the transcripts of each sample was approximately 44%. The transcripts obtained by Trinity splicing were used as reference sequences for subsequent analysis. The longest cluster sequence was obtained by corset hierarchical clustering for subsequent analysis. The lengths of the transcripts and cluster sequences were statistically analyzed. The results are shown in Appendix A. In this sequencing, the total number of transcripts was 406,864. The number of unigenes was 328,951. The average lengths of the transcripts and unigenes were 921 and 1073 bp, respectively, and the N50 values were 1426 and 1517 bp, respectively. These indicators indicate that the data obtained by sequencing were of high quality, ensuring the reliability of subsequent analyses.

#### 3.2.2. Functional Annotation and Classification of Unigenes

To obtain comprehensive gene function information, a blaxtx comparison was conducted between Unigenes and seven functional databases (including NR, TrEMBL, Pfam, KOG/cog, Swiss-Prot, KEGG, and GO). The annotation results are shown in Appendix A. All unigenes were annotated in the seven databases. The number of genes annotated successfully in at least one database accounted for 46.23%. The proportion of genes annotated in the NR database was the largest (45.57%), and that in the KOG/COG database was the smallest (28.24%). The species distribution statistics of unigenes annotated by NR (Appendix A) were determined. All five species annotated by NR belonged to Monocotyledonae, and the first three species, *Asparagus officinalis* (42.44%), *Elaeis guineensis* (5.38%), and *Phoenix dactylifera* (4.54%), belonged to Asparagiaceae and Palmaceae (originating from Liliaceae), respectively. The unigene annotation information of the garlic transcriptome under low-temperature stress had high reliability, providing a basis for a more accurate study of gene function. In addition, 30.47% of unigenes have not yet been annotated in other species. It is speculated that these unigenes have not been sequenced or are specific to garlic.

#### 3.2.3. Identification of DEGs 

To identify the DEGs, gene expression levels were estimated with the fragments per kilobase of exon per million fragments mapped (FPKM) values. Under low-temperature stress, |log2fold change| ≥ 1 and FDR < 0.05 were selected as the screening conditions for DEGs. The DEGs were calculated relative to the gene expression amount of each sample under the control conditions. In this study, the transcriptional profiles of TDT1 (low-temperature stress treatment for 1 d), TDT5 (low-temperature stress treatment for 5 d), and TCK (control group without low-temperature stress treatment) were analyzed (Figure 2). The number of DEGs differed at different times of low-temperature stress. Compared with TCK, the longer the stress time, the greater the total number of DEGs. More upregulated and downregulated genes (compared with TCK) were produced, indicating that a large number of genes were induced or inhibited under low-temperature stress. This change causes plants to resist stress or leads to metabolic imbalance.

#### 3.2.4. KEGG Enrichment Analysis of Differentially Expressed Carbohydrate Genes Produced by Low-Temperature Stress

To further understand the function of carbohydrate differential expression genes involved in response to low-temperature stress and their main signal pathways, KEGG pathway analysis was conducted to screen the DEGs (Figure 3). For TCK_vs_TDT5 (compared to the untreated control group on the fifth day of low-temperature stress), the starch and sucrose metabolism (ko00500) pathways were enriched with the largest number of DEGs followed by the glycolysis/gluconeogenesis (ko00010) and amino sugar and nucleotide sugar metabolism (ko00520) pathways, but the number of DEGs in these two pathways was equal in TCK_vs_TDT1 (compared to the untreated control group on the first day of low-temperature stress) and TCK_vs_TDT5. Among these, the same number of DEGs in the two periods were associated with galactose metabolism (ko00052) and fructose and mannose metabolism (ko00051). In addition, for TCK_vs_TDT5, the metabolism of starch and sucrose (ko00500), pentose phosphate pathway (ko00030), mutual transformation of pentose and glucuronide (ko00040), degradation of other glycans (ko00511), glycosylphosphatidylinositol (GPI)-anchored biosynthesis (ko00563), sphingolipid biosynthesis ganglion series (ko00604), and glycosaminoglycan degradation (ko00531) were annotated in TDT5. The number of differential expression genes was more than that of TCK_vs_TDT1, and other types of O-glycine biosynthesis (ko00514) and N-sugar chain biosynthesis (ko00510) were only found in TCK_vs_TDT5. These results indicate that the number of DEGs in different carbohydrate metabolism pathways increased with the extension of stress time.

#### 3.2.5. qRT–PCR Verification

To confirm the accuracy and repeatability of the RNA-seq results, we randomly screened 10 key genes of carbohydrate metabolism and performed qRT–PCR validation. The expression patterns of six genes were consistent with the changes in FPKM values (Figure 4). These results indicate the reliability of the RNA-seq data.

### 3.3. Metabolite Analysis Results

#### 3.3.1. Detection of Metabolites and Screening of Differential Expression Metabolites

After exposure to low-temperature stress, the metabolites of garlic seedlings changed significantly. A total of 795 metabolites (Appendix A) were detected in MCK (control group without low-temperature stress treatment), MDT1 (low-temperature stress treatment for 1 d), and MDT5 (low-temperature stress treatment for 5 d). Among them, 626 metabolites did not respond to cold stress (Appendix A). Screening conditions of fold change ≤ 0.5 and VIP ≥ 1 were selected to identify differential expression metabolites. Compared with the control (MCK), 37 metabolites accumulated differently after 1 d of low-temperature treatment (MDT1), of which 17 metabolites increased, and 20 decreased. The main metabolites mainly included organic acids and derivatives, lipids, alkaloids, flavonoids, phenylpropanoid, amino acids and derivatives, and vitamins and derivatives. Compared with the control (MCK), 158 metabolites accumulated differently after 5 d of low-temperature treatment (MDT5), among which 95 metabolites increased, and 63 decreased. Compared with the control (MCK), nucleotides and their derivatives, phenols, indoles and their derivatives, alcohols, and sugars increased (Figure 5).

The number of differential expression metabolites and specific accumulated metabolites detected after 1 and 5 d of low-temperature treatment were visualized using a Venn diagram. According to Figure 6, 26 differential expression metabolites (Appendix A) co-existed at 1 and 5 d of low-temperature stress, mainly involving lipids, alkaloids, flavonoids, organic acids and derivatives, phenylpropanoid, amino acids and derivatives, and anthocyanin-related metabolites. There were 11 differential expression metabolites that only existed at 1 d of low-temperature stress (compared with MCK), and 132 of them were specific to 5 d (compared with MCK). The accumulation of an increasing number of metabolites changed with the extension of the low-temperature stress time, and the difference was obvious among the treatment groups.

#### 3.3.2. KEGG Enrichment Analysis of Differential Expression Metabolites under Low-Temperature Stress

To further understand the function of the differential expression metabolites involved in the response to low-temperature stress and the main signaling pathways, KEGG enrichment analysis was carried out for the selected differential expression metabolites. The enrichment results are shown in Figure 7. After low-temperature treatment, the pathways involved in differential expression metabolites mainly included metabolism, environmental information processing, and genetic information processing. Among them, only metabolism-related pathways were involved after 1 d of low-temperature stress, while the above three pathways were involved after 5 d of low-temperature stress. At the same time, there were 12 pathways involved in the differential accumulation of metabolites after 1 d of low-temperature stress (compared with MCK), namely phenylpropionic acid biosynthesis (ko00940), stilbenes, diarylheptanes and gingerols (ko00945), secondary metabolites (ko01110), flavonoid biosynthesis (ko00941), isoflavone biosynthesis (ko00943), flavonoids and flavonol biosynthesis (ko00944), metabolic pathway (ko01100), tyrosine metabolism (ko00350), nicotinic acid and nicotinamide metabolism (ko00760), steroid biosynthesis (ko00100), anthocyanin biosynthesis (ko00942), and ubiquinone and other terpenoid quinone biosynthesis (ko00130) were changed, and the pathway related to glucose metabolism was not noted. After 5 d of low-temperature stress (compared with MCK), the following pathways were prevalent: purine metabolism (ko00230), caffeine metabolism (ko00232), metabolic pathway (ko01100), secondary metabolite biosynthesis (ko01110), phenylpropionic acid biosynthesis (ko00940), folic acid synthesis (ko00790), nicotinic acid and nicotinamide metabolism (ko00760), tryptophan metabolism (ko00380), flavonoid and flavonol biosynthesis (ko00944), starch and sucrose metabolism (ko00) 500), arginine and proline metabolism (ko00330), carbapenem biosynthesis (ko00332), aminoacyl tRNA biosynthesis (ko00970), amino acid biosynthesis (ko01230), ABC transporter (ko02010), cysteine and methionine metabolism (ko00270), pyrimidine metabolism (ko00240), flavonoid biosynthesis (ko00941), tyrosine metabolism ACB (ko00350), butylene biosynthesis acid metabolism (ko00650), stilbenes, diarylheptanes and gingerols biosynthesis (ko00945), isoflavone biosynthesis (ko00943), galactose metabolism (ko00052), ascorbic acid and uronic acid metabolism (ko00053), inositol phosphate metabolism (ko00562), phosphatidylinositol signaling system (ko04070), pantothenic acid and coenzyme A biosynthesis (ko00770), vitamin B6 metabolism (ko00750), phenylalanine metabolism (ko00360), glutathione metabolism (ko00480), isoquinoline alkaloid biosynthesis (ko00950), and anthocyanin biosynthesis (ko00942). The results showed that soluble sugars began to appear after garlic was exposed to low-temperature stress for 5 d.

#### 3.3.3. Screening of Differential Expression Carbohydrate Metabolites

Garlic seedlings can resist stress by regulating the accumulation of carbohydrate metabolites after low-temperature treatment. As shown in Table 1, 20 carbohydrate-related differential expression metabolites, including D(-)-Threose (pma0134), D(+)-Melezitose O-rhamnoside (pmb2653), maltotetraose (pmb2858), D-(+)-Sucrose (pme0519), DL-Arabinose (pme2019), D-Sedoheptuiose 7-phosphate (pme3163), galactinol (pmf0032), and glucose-1-phosphate (pmf0035), were identified. Ten carbohydrate metabolites, including D-Fructose 6-phosphate-disodium salt (pmf0220) and Sucralose (pmf0574), were not reported in the KEGG database. The other 10 carbohydrate metabolites involved 13 KEGG metabolic pathways: glycolysis/Gluconeogenesis (ko00010), pentose phosphate pathway (ko00030), galactose metabolism (ko00052), starch and sucrose metabolism (ko00500), amino sugar and nucleotide sugar metabolism (ko00520), indole alkaloid biosynthesis (ko00901), metabolic pathways (ko01100), biosynthesis of secondary metabolites (ko01110), carbon metabolism (ko01200), fructose and mannose metabolism (ko00051), inositol phosphate metabolism (ko00562), biosynthesis of various plant secondary metabolites (ko00999), and carbon fixation in photosynthetic organisms (ko00710).

### 3.4. Correlation Analysis of the Transcriptiome and Metabolome

#### 3.4.1. KEGG Enrichment Analysis of Differential Expression Genes and Differential Expression Metabolites

To understand the relationship between genes and metabolites, the DEGs and differential expression metabolites of the same group were mapped together to the KEGG pathway map. To better understand the enrichment degree of DEGs and metabolites in each pathway, we drew a histogram based on the enrichment results of differential expression metabolites and DEGs, showing their correlation pathways. As shown in Figure 8, except for a few pathways, the enrichment degree of differential expression genes was generally higher than that of differential expression metabolites, regardless of 1 or 5 d of exposure to low-temperature stress. Under low-temperature stress, there were 33 interaction pathways between DEGs and metabolites, among which only 12 gene metabolite-related pathways were found after 1 d of low-temperature treatment, and 31 pathways were increased after 5 d of low-temperature treatment. KEGG enrichment analysis of metabolites showed that the pathway related to sugar metabolism was not annotated after 1 d of low-temperature stress, while starch and sucrose metabolism and galactose metabolism were annotated when the stress time reached 5 d.

#### 3.4.2. Weighted Gene Co-Expression Network Analysis

To understand the correlation between the sample expression matrix and the sample carbohydrate metabolites, WGCNA was conducted to study the co-expression network of DEGs. Based on their similar expression patterns, a total of five co-expression modules were identified (Figure 9A), namely gene matrices of different modules, such as turquoise, brown, green, blue, and yellow. In addition, this study also created a correlation heatmap between modules and carbohydrate metabolites (Figure 9B), which showed a high correlation between the gene matrices of the turquoise and blue modules and the sample carbohydrate metabolites (|r| ≥ 0.8). Among them, the genes of the blue module were positively correlated with trehalose 6-phosphate and maltotetraose in carbohydrate metabolites (r > 0.8) and negatively correlated with D-(+)-Mannose (r = −0.84). The turquoise module gene was positively correlated with DL-Arabinose, D(+)-Glucose, D-(+)-Galactose, and D-(+)-Mannose in carbohydrate metabolites (r > 0.8), and negatively correlated with galactinol (r = −0.81).

Based on the above research, we screened key enzyme genes related to fructan metabolism in the blue and turquoise modules, and the screening results are shown in Table 2. We found three genes related to fructan metabolism: sucrose: sucrose-1-fructosyltransferase (*1-SST*); fructan: fructan 6G-fructosyltransferase gene (*6G-FFT*); and fructan 1-exohydrolase gene (*1-FEH*). The ID number of the four transcripts of *1-SST* in the KEGG notes was K21351 (sucrose: sucrose fructosyltransferase [EC:2.4.1.99]). The ID number of the two transcripts of *6G-FFT* in the KEGG notes was K21352 (6(G)-fructosyltransferase [EC:2.4.1.243]). The ID number of the six transcripts of *1-FEH* in the KEGG note was K20848 (fructan beta-(2,1)-fructosidase [EC:3.2.1.153]). The above KEGG note results indicate that the screening results for the fructan gene in this study are reliable. Under low-temperature stress, the expression of the garlic fructan transferase gene (*1-SST* and *6G-FFT*) was upregulated, and the extent of upregulation increased with the extension of stress time (compared with TCK). However, the expression of the fructan hydrolase gene (*1-FEH*) was both upregulated and downregulated, but mainly upregulated (compared with TCK).

#### 3.4.3. Correlation Analysis and Hub Genes Screening of Key Genes in Fructan Metabolism and Carbohydrate Metabolites

To clarify the mechanism of garlic fructan metabolism in response to low temperatures, a correlation network analysis was conducted on three key enzyme genes for fructan metabolism obtained from WGCNA analysis (Table 2). A total of 12 transcripts and eight carbohydrate differential expression metabolites were identified (Figure 10). Finally, 10 transcripts were screened and found to have a correlation greater than 0.8 with carbohydrate differential expression metabolites. It is speculated that the expression of key enzyme genes in fructan metabolism directly or indirectly affects the metabolic process of these carbohydrate metabolites.

The key genes of fructan metabolism were positively correlated with the seven carbohydrate metabolites. That is, the upregulated (compared with TCK) expression of the key genes of fructan metabolism promoted the accumulation of carbohydrate metabolites associated with them. In contrast, it inhibited the accumulation of carbohydrate metabolites. According to Table 2, except for the downregulation of Cluster-4573.153574 (*1-FEH*), the transcript expression of other fructosyltransferase genes (*1-SST* and *6G-FFT*) and exohydrolase genes (*1-FEH*) showed an upward trend (compared with TCK). As shown in Figure 10, DL-Arabinose (pme2019), D(+)-Glucose (pme1846), and D-(+)-Galactose (pmf0139) were only affected by the downregulation (compared with TCK) of Cluster-4573.153574 (*1-FEH)*, thus reducing the accumulation of these three carbohydrate metabolites. The transcript expression of the other nine key genes for fructan metabolism was upregulated (compared with TCK), which increased the accumulation of four carbohydrate metabolites, including maltotetraose (pmb2858), D-(+)-Sucrose (pme0519), trehalose 6-phosphate (pmb3088), and galactinol (pmf0032). In addition, the fructosyltransferase genes (*1-SST* and *6G-FFT*) had a negative correlation with D-(+)-Mannose (pmf0138). The upregulation of the fructosyltransferase gene did not promote the accumulation of more mannose but may have reduced its accumulation. Different from the fructosyltransferase gene, Cluster-4573.153574 (*1-FEH*) had a positive correlation with D-(+)-Mannose (pmf0138), and the decrease in mannose content may have been affected by the downregulation of Cluster-4573.153574 (*1-FEH*) expression. Trehalose 6-phosphate (pmb3088) was associated with the most transcripts, including eight transcripts of three key enzyme genes for fructan metabolism. Trehalose 6-phosphate was upregulated in DT1 and DT5 (Table 1) and was also the only carbohydrate metabolite with upregulated content during these two treatment periods. The transcript expression level of the key enzyme genes associated with fructan metabolism also increased with the prolongation of treatment time. This was one of the key findings of this study.

In general, different transcripts of a gene can regulate multiple metabolites, and one metabolite is affected by the expression of multiple transcripts. When the garlic was subjected to low-temperature stress, it first induced or inhibited the changes in key genes of garlic fructan metabolism to change the downstream metabolites and finally affected fructan metabolism.

To screen the hub genes of the key enzyme genes of fructan metabolism, we screened 10 transcripts related to fructan metabolism based on correlation analysis (Figure 10). Combined with WGCNA analysis, we calculated the connectivity of each transcript in the module (in the same module, the genes with larger kWithin values were considered hub genes). As shown in Table 3, except for Cluster-4573.153574, located in the turquoise module, the other nine transcripts related to fructan metabolism were all located in the blue module. In the blue module, according to the numerical value of kWithin, Cluster-4573.161559 (*6G-FFT*) was the hub gene; in the turquoise module, Cluster-4573.153574 (*1-FEH*) served as the hub gene. The hub genes obtained from the above two screenings will be of great significance for the future in-depth work of this study.

#### 3.4.4. Analysis of Sugar Metabolism Pathway under Low-Temperature Stress

To further study the regulation and adaptation mechanism of sugar metabolism in garlic under low-temperature stress, combined with the analysis results of the KEGG pathway related to sugar metabolism in the transcriptome and metabolome, different carbohydrate metabolism pathway maps were constructed (Figure 11). The results showed, in detail, the carbohydrate metabolism pathway of garlic under low-temperature stress, in which the source and metabolic pathways of carbohydrate metabolites can be clearly understood.

As shown in Figure 11, starch in garlic chloroplasts decomposes under low-temperature stress, and sucrose is first formed in the cytoplasm through a certain transport pathway. Sucrose in the cytoplasm decomposes into fructose and glucose under the action of cytoplasmic sucrose invertase (CINV). The expression of the *CINV* gene increased with the extension of stress time, which made sucrose in the cytoplasm decompose more fructose and glucose, providing more substrates for the synthesis of carbohydrate metabolites. Under the action of the sucrose transporter (SUT), sucrose in the cytoplasm is transported to the vacuole or extracellularly to participate in other glucose metabolism pathways as a substrate of glucose metabolism. In this study, the *SUT* gene was significantly upregulated in garlic under low-temperature stress for 5 d, indicating that more sucrose was transported to vacuoles and extracellularly. Some of the sucrose transported to the vacuole is decomposed into glucose and fructose under the action of vacuolar invertase (vINV), but under low-temperature stress, the gene is downregulated, indicating that most sucrose may be used as the substrate for synthesis. In the fructan synthesis pathway, the expression of *1-SST* and *6G-FFT* were upregulated and increased with the extension of stress time. However, there was no difference in *1-FFT* expression in the fructan synthesis pathway, suggesting that the gene was not activated by low-temperature stress. In the fructan degradation pathway, *1-FEH* expression showed an upward and downward trend, but the upward trend was more obvious. The above results indicate that fructan metabolism is a process of synthesis during hydrolysis under low-temperature stress and that the upregulation of the synthetase gene is greater, which indicates that there may be more fructan accumulation at this stage.

In addition, as shown in Figure 10, trehalose 6-phosphate, the carbohydrate metabolite most associated with the fructan gene, showed an upward trend in content under low-temperature stress. Interestingly, the expression level of trehalose 6-phosphate metabolism-related genes (*TPS*, Trehalose phosphate synthase) showed a downward trend (Figure 11), but their content showed an upward trend during the stress process (Figure 10). It is speculated that the upregulation of fructan metabolism-related genes plays a positive promoting role in trehalose 6-phosphate accumulation.

## 4. Discussion

Low-temperature stress is one of the main environmental stressors with which many plants must cope. In the long evolutionary process, plants have formed a complex system to sense and respond to low-temperature stress [22]. Sugar metabolism has been proven to play a key role in the overwintering response of plants to cold stress [23,24,25]. Previous studies have shown that sucrose metabolism is closely related to the cold tolerance of plants and coordinated with an increase in sucrose [26,27]. Fructan is a fructose polymer synthesized from the sucrose in plant vacuoles [28]. Under abiotic stress, the fructan content in plants increases and combines with the cell membrane to ensure cell morphology, reduce cell damage, and enhance plant stress resistance [29,30]. In addition, fructans can reduce oxidative stress and photoinhibition caused by low temperatures [31,32]. However, the mechanism of fructan resistance induced by low-temperature stress in garlic remains unclear. 

To explore the low-temperature-induced fructan-related genes and their mechanisms in garlic, the transcriptome library of low-temperature- and normal-temperature-treated garlic was constructed by high-throughput sequencing technology, and metabolites were identified by ultra-performance liquid chromatography–tandem mass spectrometry (UPLC–MS/MS). The effective number of unigenes obtained from garlic leaf transcriptome sequencing was 328,951. According to Figure 2, with the extension of stress time, the greater the number of DEGs. This result is consistent with Fu et al. [11] and Lee et al. [33]. To further understand the function and metabolic pathway of carbohydrate DEGs responding to low-temperature stress, the analysis of DEGs in the carbohydrate KEGG pathway showed that the starch and sucrose metabolism (ko00500) pathway had the largest number of DEGs, indicating that this pathway plays an important role in regulating low-temperature stress in garlic seedlings. Metabonomic research has been proven to be helpful in understanding the mechanism of genotype–environment interaction by comparing the transcriptome of the control genotype, stress treatment, and development stage [34]. Using metabolomics technology, 795 metabolites in garlic were identified, involving 44 KEGG metabolic pathways. By analyzing the KEGG pathway of the screened differential expression metabolites, the pathway related to sugar metabolism could only be annotated at the late stage of stress, which further indicated that long-term low-temperature stress induced the accumulation of carbohydrate metabolites. The above transcriptomic and metabonomic data provide an important database for further exploring the low-temperature metabolic pathway in garlic.

Research has shown that the fructans in garlic belong to the new series of inulin fructans [35,36], and the synthase genes involved in its metabolic pathway were sucrose: sucrose 1-fructosyltransferase genes (*1-SST*), fructan: fructan 6G fructosyltransferase genes (*6G-FFT*), and fructan: fructan 1-fructosyltransferase genes (*1-FFT*). Fructan 1-exohydrolase genes (*1-FEH*) are degrading enzymes. However, in this study, fructan: fructan 1-fructosyltransferase (*1-FFT*) was not activated, which may be due to its high energy demand. We created a correlation heat map between gene modules and carbohydrate metabolites through WGCNA analysis (Figure 9) and screened 12 gene transcripts related to fructan metabolism in key modules (blue and turquoise module). Based on the correlation analysis between the selected fructan metabolism-related genes and carbohydrate metabolites, we constructed a network diagram (Figure 11) and demonstrated that the key genes of fructan metabolism were positively correlated with seven carbohydrate metabolites and negatively correlated with one carbohydrate metabolite. Among the transcripts of the key genes of fructan metabolism, except for the downregulation of Cluster-4573.153574 (*1-FEH*), the expression of other transcripts was upregulated (compared with TCK), and the accumulation of carbohydrate metabolites that had a positive correlation with the transcripts with the upregulated (compared with TCK) expression increased with the extension of stress time. Moreover, due to the downregulation of the expression of Cluster-4573.153,574 (*1-FEH*), the accumulation of carbohydrate metabolites was positively related to its decrease. In addition, in this network diagram, only the accumulation of D-(+)-Mannose (pmf0138) did not increase due to the upregulation (compared with TCK) of fructosyltransferase genes (*1-SST* and *6G-FFT*). However, it may have been affected by the downregulation of the expression of Cluster-4573.153574 (*1-FEH*), and its accumulation showed a downward trend. Based on the results, it is speculated that the expression of key genes of fructan metabolism in garlic under low-temperature stress affects the content of metabolites in the sugar metabolism pathway. The accumulation of carbohydrate metabolites may be jointly affected by fructosyltransferase genes (*1-SST* and *6G-FFT*) and exohydrolase genes (*1-FEH*) or may be affected by only one of the enzyme genes. Furthermore, according to the types of key genes of fructan metabolism associated with carbohydrate metabolites whose metabolite accumulation increased, a single type of fructosyltransferase gene (*1-SST* and *6G-FFT*) may directly induce the increase of the accumulation of carbohydrate metabolites associated with it (such as pmf0032), but the increase of carbohydrate metabolite accumulation associated with it may not be induced by the increase of the exohydrolase gene (*1-FEH*) alone. However, it can only induce an increase in metabolite accumulation together with the upregulated (compared with TCK) fructosyltransferase genes (*1-SST* and *6G-FFT*) (such as pmb2858 and pmb3088). In contrast, the downregulation (compared with TCK) of the exohydrolase gene (*1-FEH*) may lead to a decrease in the accumulation of carbohydrate metabolites associated with it (such as pmf0138, pme2019, pme1846, and pmf0139). In addition to the above results, two hub genes, Cluster-4573.161559 (*6G-FFT*) and Cluster-4573.153574 (*1-FEH*), were screened based on the kWithin values of each transcript in the gene module. This will lay the foundation for future research on the function of key enzyme genes in garlic fructan metabolism.

To adapt to abiotic stress, plants accumulate photosynthate products in the form of fructan rather than starch in cell vacuoles [37]. Sucrose is considered a key integrated regulatory molecule that controls the expression of genes related to plant metabolism, stress resistance, and growth and development [38,39]. In this study (Figure 11), sucrose transport in cells cannot be separated from the role of sucrose transporters (SUTs). Under low-temperature stress, the *SUT* gene is upregulated (compared with TCK), which promotes sucrose transport [40] and makes the intracellular sugar metabolism pathway go smoothly. Sucrose in vacuoles is synthesized into glycan under the catalysis of fructosyltransferase, and a portion of sucrose in vacuoles is decomposed into fructose and glucose by the vacuolar invertase gene (*vINV*). Fructosyltransferase genes, vacuolar invertase genes, and cell wall invertase genes (*cwINV*) in vacuoles belong to the 32 families of glycosyl hydrolases [41,42]. Under low-temperature stress, the gene expression of fructosyltransferase genes (*1-SST* and *6G-FFT*) was upregulated (compared with TCK), while the expression of *vINV* was downregulated. It is speculated that most of the sucrose in the vacuole is used as the substrate for fructan synthesis, which can better resist low temperatures. In addition, sucrose transported to the extracellular space was decomposed under the effect of upregulated (compared with TCK) *cwINV*, indicating that more fructose and glucose might be formed in the extracellular space to resist the low-temperature environment. As shown in Figure 10, the fructosyltransferase gene (*1-SST* and *6G-FFT*) and the exohydrolase gene (*1-FEH*) were positively correlated with trehalose 6-phosphate (pmb3088), and the amount of fructan metabolism genes and trehalose 6-phosphate metabolites involved was upregulated after low-temperature stress in garlic. Combined with the research results in Figure 11, after exposure to cold stress, the expression of the trehalose-6-phosphate synthase gene (*TPS*) in garlic, the upstream gene of trehalose-6-phosphate (pmb3088), did not increase and had a downward trend, but the accumulation of trehalose-6-phosphate showed a significant upward trend in the early and late stages of stress. Therefore, it is speculated that the increase in trehalose 6-phosphate accumulation is mainly affected by the key genes associated with fructan metabolism. This further shows that the expression of key genes of fructan metabolism under low-temperature stress can not only improve the accumulation of its own metabolites but also positively promote the accumulation of metabolites in the sugar metabolism pathway. It is of great significance for garlic to better resist a low-temperature environment. The results of this study need to be verified in future research.

## 5. Conclusions

In this study, we used metabolome and transcriptome analyses to identify key genes of fructan metabolism in garlic under low-temperature stress, and *1-SST*, *6G-FFT* and *1-FEH* were screened. Metabolite analysis results demonstrated that garlic seedlings resist low-temperature stress by regulating the accumulation of carbohydrate metabolites. Through constructing the correlation network diagram between the key genes of fructan metabolism and carbohydrate metabolites, we found that fructan transferase genes (*1-SST* and *6G-FFT*) had an obviously positive role in the response of garlic fructan to low temperatures. Through WGCNA analysis, we predicted the hub genes of the key enzyme genes of fructan metabolism, which also laid the foundation for future functional verification of the key enzyme genes of garlic fructan metabolism. The results of this study more intuitively show the sugar metabolism pathway and its expression characteristics in garlic leaf cells under low-temperature stress and provide valuable information for future research. The research results of this study will provide an important reference value for us to clarify the metabolic pathways of garlic fructan in the future. In future work, we will also carry out functional verification of the hub genes Cluster-4573.161559 (*6G-FFT*) and Cluster-4573.153574 (*1-FEH*) screened in this paper to better understand the regulatory mechanism of garlic fructan-regulating abiotic stress, such as low-temperature stress.

## Figures and Tables

**Figure 1 genes-14-01290-f001:**
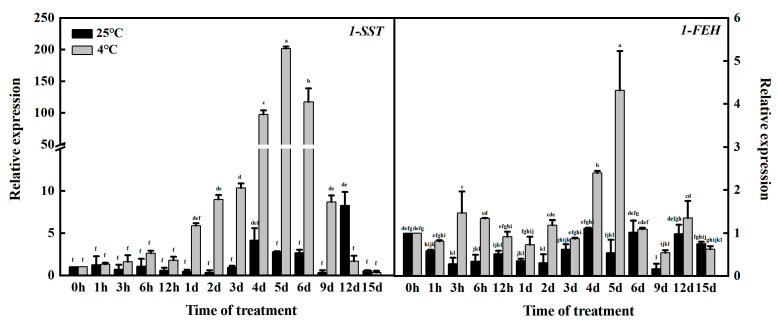
Expression of *1-SST* and *1-FEH* at different stages of low-temperature stress. Different letters in the same panel indicate statistical significance (*p* < 0.05).

**Figure 2 genes-14-01290-f002:**
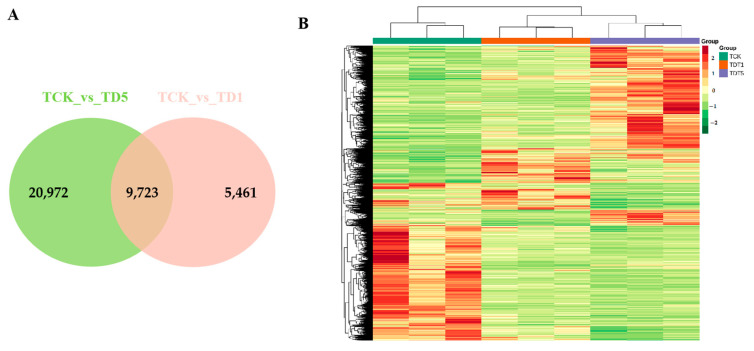
Differential expression gene analysis map. (**A**) Venn diagram of the differentially expressed genes identified in the two treatments. (**B**) Transcriptional heat map of TDT1, TDT5, and TCK.

**Figure 3 genes-14-01290-f003:**
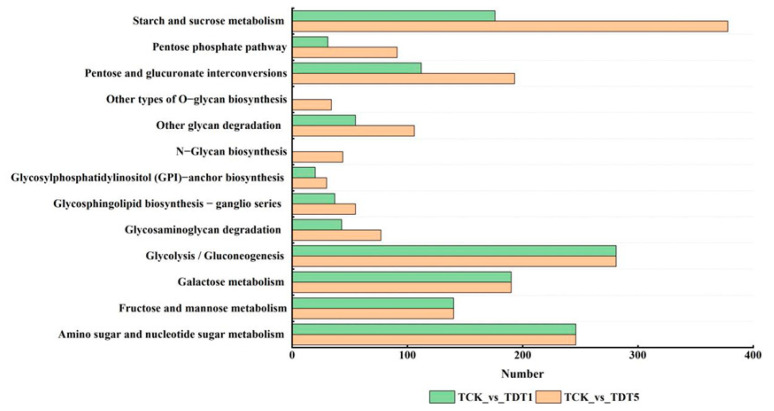
KEGG enrichment analysis of differentially expressed genes.

**Figure 4 genes-14-01290-f004:**
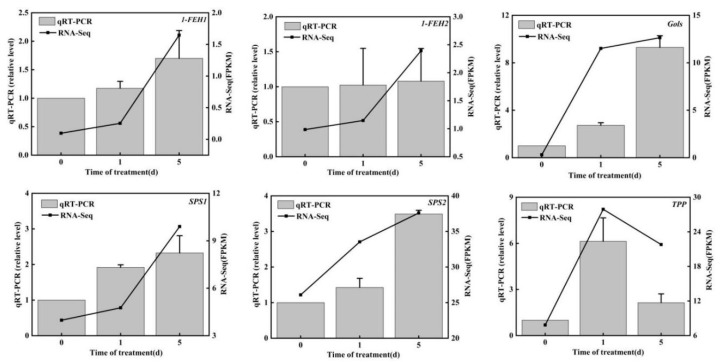
qRT–PCR verification of 10 key genes of carbohydrate metabolism.

**Figure 5 genes-14-01290-f005:**
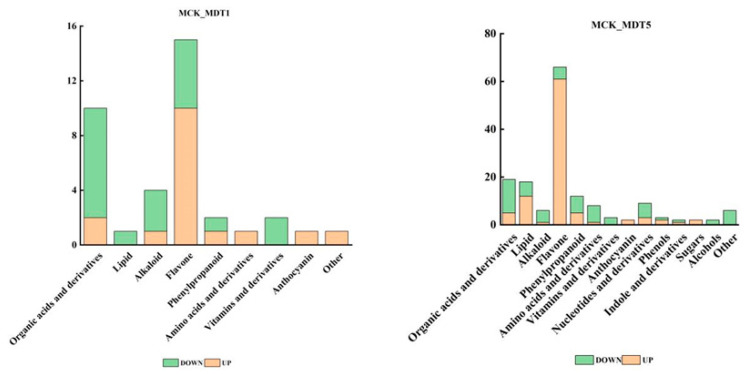
Different types of differential expression metabolites under different treatments.

**Figure 6 genes-14-01290-f006:**
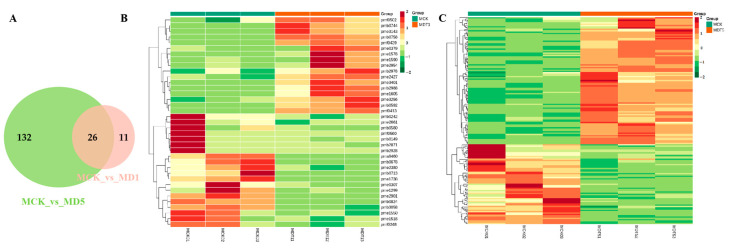
Differential expression metabolite analysis graph. (**A**) Venn diagram of differential expression metabolites. (**B**) Metabolic heat map of MCK and MDT1. (**C**) Metabolic heat map of MCK and MDT5.

**Figure 7 genes-14-01290-f007:**
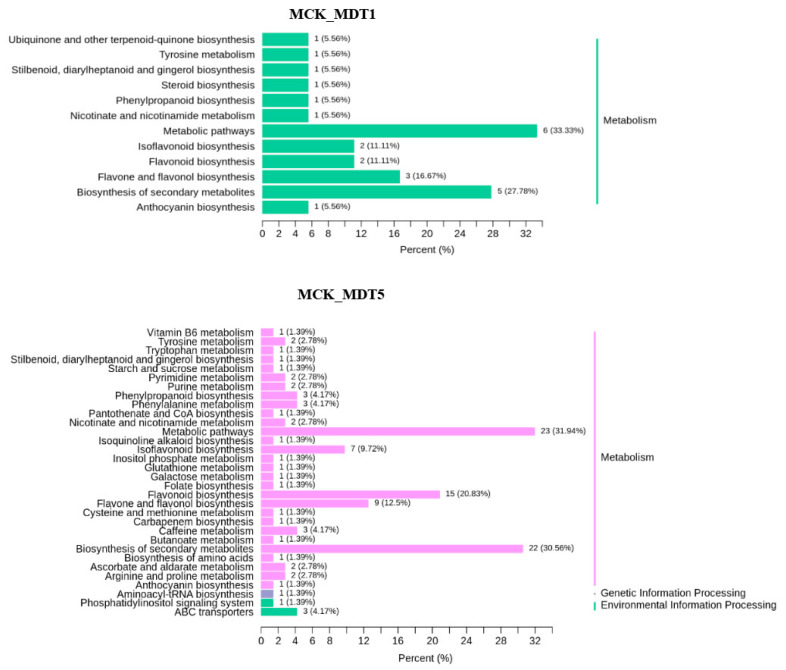
Differential expression metabolite KEGG classification map. MCK_vs_MDT1: Compared to the untreated control group on the first day of low-temperature stress; MCK_vs_MDT5: Compared to the untreated control group on the fifth day of low-temperature stress.

**Figure 8 genes-14-01290-f008:**
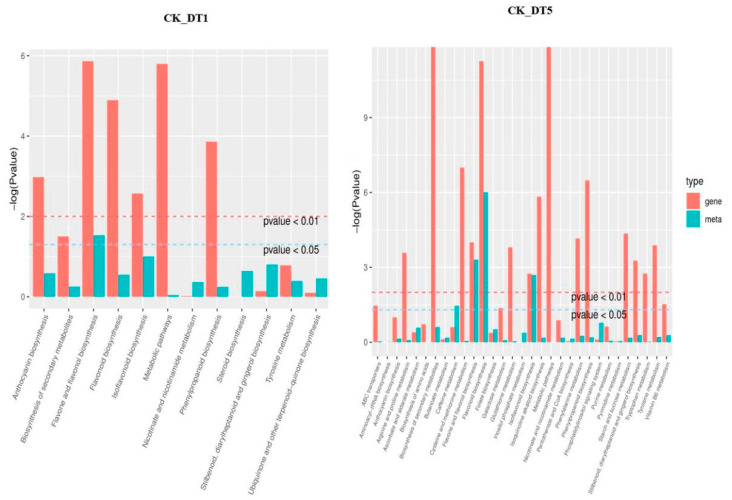
KEGG enrichment analysis *p*-value histogram. Gene represents differentially expressed genes; meta represents differential expression metabolites; CK_vs_DT1: Compared to the untreated control group on the first day of low-temperature stress; CK_vs_DT5: Compared to the untreated control group on the fifth day of low-temperature stress.

**Figure 9 genes-14-01290-f009:**
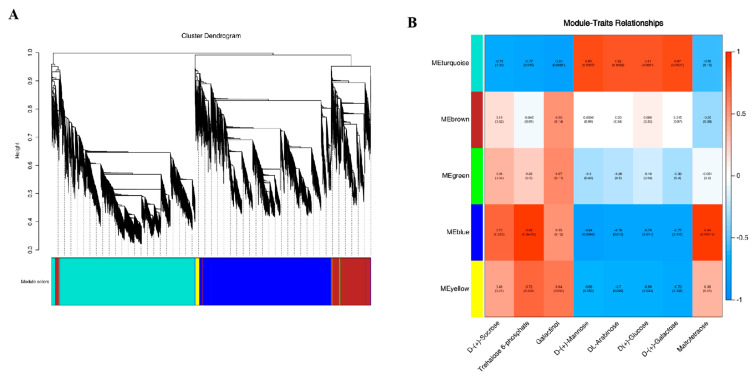
Transcriptomic and metabolomic correlation analyses. (**A**) Module hierarchical clustering tree graph. (**B**) Correlation between gene modules and carbohydrate metabolites.

**Figure 10 genes-14-01290-f010:**
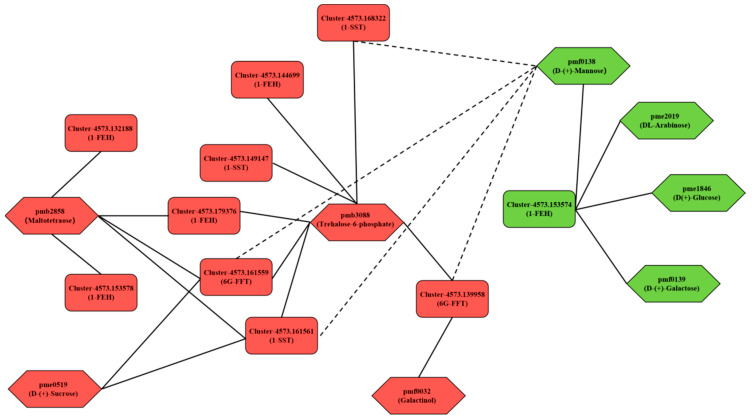
Correlation network of key genes of fructan metabolism and carbohydrate metabolites under low temperatures. The solid line represents a positive correlation, and the dotted line represents a negative correlation. Red indicates upregulated expression, and green indicates downregulated expression.

**Figure 11 genes-14-01290-f011:**
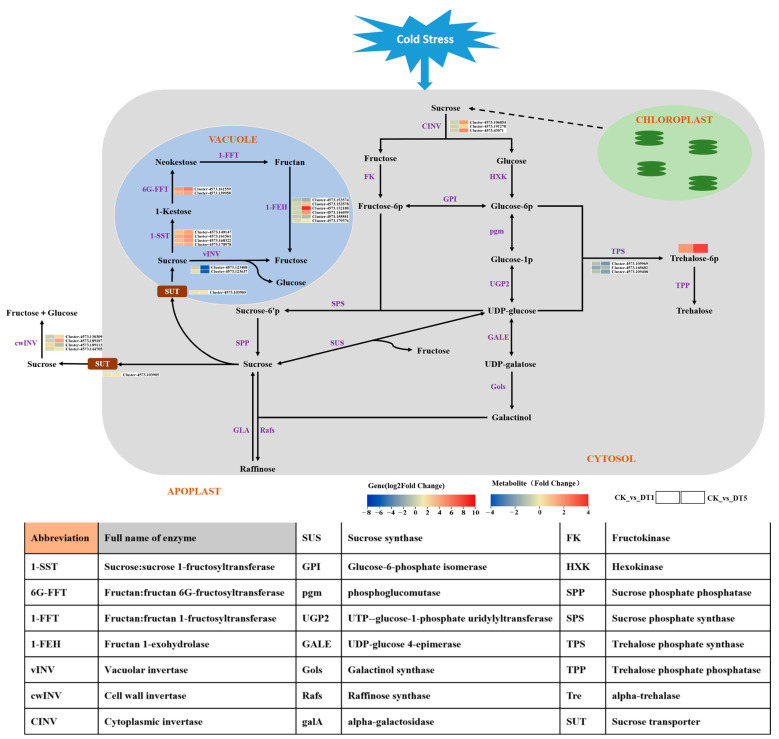
Analysis of the garlic sugar metabolic pathway under low temperatures.

**Table 1 genes-14-01290-t001:** Statistics of Sugar Metabolites.

Index	Compounds	ko_ID	Regulated	
			MCK_vs_MDT1	MCK_vs_MDT5
pma0134	D(-)-Threose	-	-	-
pmb0786	Glucosamine	ko00520, ko01100	-	-
pmb2653	D(+)-Melezitose O-rhamnoside	-	-	-
pmb2858	Maltotetraose	-	-	up
pmb3088	Trehalose 6-phosphate	ko00500, ko01100	up	up
pme0519	D-(+)-Sucrose	-	-	up
pme1684	D-(+)-Glucono-1,5-lactone	ko00030, ko01100, ko01110, ko01200	-	-
pme1846	D(+)-Glucose	ko00010, ko00030, ko00052, ko00500, ko00520, ko00901, ko01100, ko01110	-	down
pme2019	DL-Arabinose	-	-	down
pme2755	N-Acetyl-D-glucosamine	ko00520, ko01100	-	-
pme3160	D-Glucose 6-phosphate	ko00500, ko00562, ko00999, ko01100	-	-
pme3163	D-Sedoheptuiose 7-phosphate	-	-	down
pme3313	D-Fructose 6-phosphate	ko00052, ko00500, ko00710, ko01100, ko01200	-	-
pmf0032	Galactinol	-	up	-
pmf0035	Glucose-1-phosphate	-	-	-
pmf0138	D-(+)-Mannose	ko00051, ko00052, ko00520, ko01100	-	down
pmf0139	D-(+)-Galactose	ko00052, ko01100	-	down
pmf0220	D-Fructose 6-phosphate-disodium salt	-	-	-
pmf0282	Melibiose	ko00052	-	-
pmf0574	Sucralose	-	-	-

**Table 2 genes-14-01290-t002:** Screening key information for fructans based on WGCNA analysis.

Gene Name	Gene ID	KEGG Annotation	TCK_vs_TDT1		TCK_vs_TDT5	
			Log2(FC)	Regulated	Log2(FC)	Regulated
*1-SST* (sucrose: sucrose 1-fructosyltransferase gene)	Cluster-4573.149147	K21351 sucrose:sucrose fructosyltransferase [EC:2.4.1.99]	2.870591819	-	4.205341664	up
	Cluster-4573.161561	K21351 sucrose:sucrose fructosyltransferase [EC:2.4.1.99]	3.722296577	up	4.685898781	up
	Cluster-4573.168322	K21351 sucrose:sucrose fructosyltransferase [EC:2.4.1.99]	3.194678677	up	4.483060574	up
	Cluster-4573.170978	K21351 sucrose:sucrose fructosyltransferase [EC:2.4.1.99]	3.126650449	up	3.692679624	up
*6G-FFT* (fructan: fructan 6G-fructosyltransferase gene)	Cluster-4573.161559	K21352 6(G)-fructosyltransferase [EC:2.4.1.243]	4.409936024	up	5.733601046	up
	Cluster-4573.139958	K21352 6(G)-fructosyltransferase [EC:2.4.1.243]	3.320464074	-	4.081188018	up
*1-FEH* (fructan 1-exohydrolase gene)	Cluster-4573.153574	K20848 fructan beta-(2,1)-fructosidase [EC:3.2.1.153]	−0.658755386	-	−1.628771042	down
	Cluster-4573.153578	K20848 fructan beta-(2,1)-fructosidase [EC:3.2.1.153]	-	-	2.790164693	up
	Cluster-4573.132188	K20848 fructan beta-(2,1)-fructosidase [EC:3.2.1.153]	-	-	8.74315447	up
	Cluster-4573.144699	K20848 fructan beta-(2,1)-fructosidase [EC:3.2.1.153]	-	-	4.53163043	up
	Cluster-4573.155551	K20848 fructan beta-(2,1)-fructosidase [EC:3.2.1.153]	−0.481771581	-	−1.077228392	down
	Cluster-4573.179376	K20848 fructan beta-(2,1)-fructosidase [EC:3.2.1.153]	0.484182905	-	1.386712343	up

**Table 3 genes-14-01290-t003:** Comparison of kWithin of key genes in fructan metabolism in the blue and turquoise modules.

Gene Name	Gene ID	Module Colors	kWithin
*1-SST*	Cluster-4573.149147	blue	3970.90
	Cluster-4573.161561	blue	5022.95
	Cluster-4573.168322	blue	4904.94
*6G-FFT*	Cluster-4573.161559	blue	5233.82
	Cluster-4573.139958	blue	3655.29
*1-FEH*	Cluster-4573.153578	blue	2584.97
	Cluster-4573.132188	blue	1437.04
	Cluster-4573.144699	blue	3038.59
	Cluster-4573.179376	blue	4277.31
	Cluster-4573.153574	turquoise	7742.60

## Data Availability

The dataset is available from the NCBI Short Read Archive (SRA) under accession number PRJNA961362.

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
