# Peer review of "Integrated Transcriptomics and Metabolomics Analysis of the Fructan Metabolism Response to Low-Temperature Stress in Garlic"

_genes, 2023, doi:10.3390/genes14061290_

Round 1

Reviewer 1 Report

Authors have documented few interesting results in garlic using transcriptome and metabolomic analysis. The paper is well structured. The analyses look too basic, and detailed analyses (Co-expression or WGCNA) are needed to well support the hypothesis. I have few general comments for improvement of the paper.

Suggestions,

1.     Authors need to enhance the quality of most of the figures.

2.     Future prospectus is not clearly explained.

3.     The manuscript needs revision for language and grammar.

4.     Log2 fold change >2 or <-2 is standard cutoff for selecting differentially expressed genes. Authors need to re-analyze this.

5.     Why hasn’t authors checked RIN value of RNA? It is standard for sequencing experiments.

6.    Authors hasn’t submitted the sequencing data to NCBI-SRA. It is mandatory step before publication.

7.     How normalization of reads performed?

8.     I will suggest running WGCNA for this study. You have some interesting background results. But everything looks too basic.

9.     Figure 8. What is meta?

Author Response

Dear reviewers,

Thank you very much for comments and constructive suggestions with regard to our manuscript titled “Integrated transcriptomics and metabolomics analysis of the fructan metabolism response to low-temperature stress in garlic” (Manuscript No. genes-2306801).

Those comments are very helpful for us to improve our manuscript. We have revised the manuscript thoroughly according to the reviewers’ constructive comments and useful suggestions. Please see the revised manuscript marked up using the “Track Changes” function. We appreciate Editors/Reviewers’ warm work earnestly, and hope that the corrections will meet with approval.

The following pages are our point-to-point response to the reviewers and detailed revisions made in the text. Please let us know if you need any further information. We are looking forward to hearing from you soon. Thanks again for your great help.

With our best regards,

Jie Tian

Reviewer 2 Report

In the present work, the authors made a comparison in the gene expression and the metabolite profiles between control and cold stressed garlic plants focusing mainly on fructan synthesis. The work is interesting, it definitely provides valuable new information in this topic. However, the manuscript cannot be accepted in its present form for the following reasons:

-        - First of all, I could not find the supplementary material. Maybe it was hidden somewhere in the online system, but I could not evaluate it. Papers are usually stable without it; however, for acceptance it should be seen.

-        - Abstract: the main message is missing.

-       -  Aim: it should be more clearly define in order to stress the novelty of the work. It is true that “there is a lack of research on its mechanism and regulatory network under stressed environment”; however, earlier findings should be briefly summarized in the introduction, even if they are in other species.

-        - Figure legends must be much more self-explanatory.

-        - The abbreviations, which are surely clear for the authors, must also be explained for the readers when they are mentioned first. For example, “TCK” is explained at its second mentioning, TDT never.

-      -  In the text, indicate clearly that what is upregulated compared to what, please!

-        - The text is full of typing errors. For example, line 50: two bacco should be tobacco, etc. (Nevertheless, the text is grammatically not so bad).

-        - Line: 324 “A total of 795 metabolites were detected at 0 h (MCK), 1 d (MDT1) and 5 d (MDT5) under low-temperature stress”. Where are these? You could only found 37 or 158 differentially changed metabolites during the cold treatment period. What was the overlap between these groups? According to Fig 6, there are 26 compounds. Which are these?

-        - Which are the main metabolite groups, which did not respond at all to cold?

-        - Define clearly the subtitles. For example: “3.3.2. KEGG Enrichment Analysis of Different Metabolites Under Low-temperature Stress”. “3.4.1. KEGG Enrichment Analysis” What is the difference between these subchapters?

-       - line 509- "Unlike Galactinol, the expression of Trehalose 6-phosphate metabolism-related gene (TPS) was down-regulated, but its content was up-regulated during stress”. Content of what?

-        - Line 586: “To adapt to the environment…” What environment? Cold?

-        -        lines 589, 607: “(Figure 11)”, “Combined with the research results in Figure 11” There are only 10 figures!

Author Response

(The authors gave the same response as above.)

Round 2

Reviewer 1 Report

The authors have addressed all my comments. I endorse paper for publication.